# Factors associated with the presence and intensity of ongoing symptoms in Long COVID

**Niels Brinkman[1], Teun Teunis[2], Seung Choi[3], David Ring[1]*, W. Michael Brode[4]**

**1** Department of Surgery and Perioperative Care, Dell Medical School, The University of Texas at Austin, **2** Department of Orthopedic Surgery & Department of Plastic and Reconstructive Surgery, The University of Pittsburgh, **3** The Center for Applied Psychometric Research, Educational Psychology Department, The University of Texas at Austin, **4** Department of Internal Medicine & Department of Population Health, Dell Medical School, The University of Texas at Austin

☉ All authors contributed equally to this work.
* david.ring@austin.utexas.edu

## Abstract

### Objective

Identification of modifiable factors associated with symptom intensity among people seeking care for Post-Acute Sequelae of SARS-CoV-2 infection (PASC) could help guide the development of comprehensive, whole-person care pathways to alleviate symptoms irrespective of potential underlying pathophysiologies. We aimed to better define the key contributors to PASC, and sought the factors associated with PASC symptom presence and intensity.

### Methods

In this cross-sectional study, 249 patients presenting for PASC care at a dedicated Post-COVID-19 clinic completed a standardized screening assessment prior to initial visit and evaluation by a general internist or nurse practitioner. We measured 46 symptoms based on the WHO's Global COVID-19 Clinical Platform Case Report Form for Post COVID Condition and performed a factor analysis and item response theory based 2-parameter logistic model to develop a population-based t-score to measure PASC symptom presence and intensity (PASC-SPI). A multivariable linear regression analysis was used to assess factors associated with PASC-SPI, accounting for demographics, comorbidities, COVID-19 infection duration and severity, and mental health.

### Results

Greater PASC-SPI was associated with greater symptoms of anxiety, a longer duration of COVID-19 infection, and hypercholesterolemia. Lower PASC-SPI was associated with older age, self-reported 1–3 units of alcohol per week, and self-reported clinician confirmation of COVID-19 diagnosis. Symptoms of anxiety accounted for a considerably higher proportion of variation in PASC-SPI than other variables.

**Data availability statement:** All relevant data are within the paper and its Supporting Information files.

**Funding:** The author(s) received no specific funding for this work.

**Competing interests:** Seung W. Choi receives honoraria for lectures, presentations, speakers' bureaus, manuscript writing or educational events at the Law School Admission Council, Tennessee State Department of Education, and Texas Coordinating Board of Higher Education. David Ring reports royalties from Skeletal Dynamics, a stipend as Deputy Editor for Clinical Orthopaedics and Related Research, honoraria for lectures, presentations, speakers' bureaus, or educational events at various Universities and Hospitals, payment for expert reviews from Lawyers, payment for expert review of claims to the Vaccine Injury Compensation Program from Health Services and Resource Administration and Department of Justice, consulting fees from Premier Healthcare Solutions, royalties for Up-to-date chapter from Wolters Kluwer Health, grants from National Institutes for Health for RO1 multi-site trial of mental health toolkit for optimal recovery from trauma study, consulting fees from Everus, and has stock options at MyMedicalHub, all outside the submitted work. W. Michael Brode receives consulting fees from Roon Healthcare as a paid consultant and content expert and has stock or stock options at Roon Healthcare, payment or honoraria for lectures, presentations, speakers' bureaus, manuscript writing or educational events at Texas Health and Human Services (honorarium paid to University of Texas at Austin), HNI Health Value Institute (honorarium paid directly as individual), and Society of Hospital Medicine (honorarium paid directly as individual), and support for attending meetings and/or travel from Society of Hospital Medicine for travel and lodging for the 2022 Meeting.

## Conclusion

Symptoms of anxiety were the strongest correlate of PASC-SPI, highlighting it as both a potential neuroinflammatory marker of PASC and a modifiable component of the illness. This emphasizes the need for comprehensive, whole person treatment strategies that integrate evidence-based interventions to address the multifaceted nature of PASC.

## Introduction

Post-Acute Sequelae of SARS-CoV-2 infection (PASC) is defined by the Centers for Disease Control and Prevention as symptoms that persist or develop at least 4 weeks after initial COVID-19 infection [1], and by the World Health Organization (WHO) as continuation or onset of new symptoms at least 3 months after initial infection and lasting at least 2 months without any other explanation [2]. This broad definition has corresponded with an estimated global prevalence of PASC of 43% (95%CI, 39% to 46%) at 28 days [3]. In contrast, the RECOVER trial, a large prospective trial with uninfected controls, identified 13 key symptoms of PASC, with 10% (95%CI, 8.8% to 11%) of people meeting their PASC definition at 6 months after confirmed COVID-19 infection [4].

Developing a discrete definition of PASC is difficult due to the illnesses' heterogeneity, and that the most common symptoms comprising identified are relatively nonspecific (fatigue, shortness of breath, dizziness, headache, and cognitive dysfunction [5,6]). In some cases, PASC is considered to have strong similarities to post-intensive care syndrome among patients who were hospitalized [7,8], and myalgic encephalomyelitis/ chronic fatigue syndrome (ME/CFS) for many patients who experienced mild to moderate illness [7]. The heterogenous nature of PASC yields a spectrum of clinical manifestations that may arise from multiple distinct pathophysiologies. In many cases there are symptom clusters that have led some to suggest interconnected pathophysiological mechanisms. Pathophysiologies under consideration include persistent inflammation, dysregulation of the immune system, disturbances in neurological signaling, and complications in neurovascular function with thromboinflammation [9–15]. This perspective necessitates seeing PASC as a clinical syndrome with diverse presentations, requiring comprehensive and integrated management approaches.

Considering human illness within the biopsychosocial paradigm of human illness—which conceptualizes illness as a product of pathophysiology, mindset (thoughts, feelings; mental health), and circumstances (social health)—opens several important opportunities for recovery as the search for the underlying pathophysiology continues. There is notable evidence pointing to the potential benefits of a more comprehensive approach to PASC which also addresses psychosocial aspects of health. First, symptoms of post-traumatic stress disorder (PTSD; 37%), anxiety (36%), and depression (47%) are common among patients with PASC[16]. Second, one investigation found that a pre-infection threshold level of symptoms of depression (2-item Patient Health Questionnaire score ≥3), a threshold level of symptoms of anxiety (2-item Generalized Anxiety Disorder scale ≥3), relatively greater levels of worry about COVID-19 (not specified whether this referred to infection with the virus or the pandemic in general), and higher perceived stress (top quartile of 4-item Perceived Stress Scale) were associated with more intense post-infection symptoms and interference with daily activities[17]. Third, a recent meta-analysis reported that pre-morbid symptoms of anxiety or depression are associated with seeking care for PASC along with comorbid asthma, diabetes, COPD, immunosuppression, and ischemic heart disease[18]. Fourth, receiving a diagnosis of PASC is associated with feelings of distress[19], more so than biomedical factors such as smoking, diabetes, hypertension, asthma, hypercholesterolemia, and higher BMI[17]. Finally,

even in sequelae that are more specific to PASC like pulmonary fibrosis and anosmia, there is notable variation in symptoms among individuals [7,8].

Yet, most considerations of PASC emphasize the search for treatable pathophysiology[15,20] rather than elaborate on potentially modifiable aspects of health including feelings of anxiety, worry, and despair. Addressing mental health symptoms in people who present for care of PASC is crucial, but it should not imply that the condition is a mental health disorder. Human illness is, by definition, psychosomatic. Our thoughts, emotions, and circumstances with regard to our bodies are important correlates of symptom intensity and impact. Notably, symptoms such as anxiety might be manifestations of PASC's inflammatory processes, and traditional diagnostic markers of depression—including concentration difficulties, reduced energy, and psychomotor changes—are frequently observed in PASC. No matter its origin, symptoms of anxiety and depression are important [21–29] and potentially modifiable elements in treatment strategies[30]. The purpose of this study was to identify which factors, including psychosocial aspects of health, are associated with symptom presence and intensity ascribed to PASC. A better understanding of the modifiable factors associated with symptom intensity among people seeking care for PASC could help guide the development of comprehensive, whole-person care pathways to alleviate symptoms.

Utilizing a clinical registry of patients presenting to a dedicated outpatient PASC clinic, with routine assessment of PASC symptoms and mental health measures, we performed a factor analysis to better define the key contributors to PASC and then seek the factors associated with PASC symptom presence and intensity. Specifically, we tested the null hypothesis that there are no variables independently associated with the presence and intensity of persistent symptoms after COVID-19 infection.

## Methods

### Design

In this cross-sectional study, between June 2021 and May 2022, patients with symptoms ascribed to PASC completed a standardized screening assessment prior to initial visit and evaluation by a general internist or nurse practitioner. The Institutional Review Board of Dell Medical School, the University of Texas at Austin approved the request to collect and analyze protected health information for research purposes, and they also waived the requirement for informed consent for use and disclosure of protected health information. All methods were performed in accordance with relevant guidelines and regulations of our Institutional Review Board.

### Setting

Consecutive patients presenting to a dedicated, multidisciplinary PASC clinic at an urban University Medical Center in the United States of America.

### Participants

To be eligible for services, individuals were required to be 18 years or older and be a minimum of 12 weeks from the onset of likely or confirmed COVID-19 infection. Proof of positive COVID-19 testing was not required to receive services if there was no obvious alternative etiology for the persistent symptoms. We did not record how many people were designated non-PASC and directed back to their primary care doctor or elsewhere, but we estimate this is less than 5%. This resulted in the inclusion of 249 patients.

## Outcome measures

We measured the presence of the following 46 symptoms using a comprehensive review of systems modeled on the WHO's Global COVID-19 Clinical Platform Case Report Form for Post COVID Condition[31] (S1 File). We omitted 2 items addressing symptoms of anxiety and depression a priori considering various aspects of mental health were assessed in more detail as explanatory variables. Symptoms were scored as positive for the responses "Yes, the symptom is still present" and "Yes, the symptom comes and goes", and negative for "Yes, but not present anymore," "Unknown," and un-answered responses, as indicated in the survey instructions.

## Factor analysis to establish PASC symptom presence and intensity score

To identity groupings of symptoms that address PASC in the WHO's Global COVID-19 Clinical Platform Case Report Form for Post COVID Condition, we performed a factor analysis and item response theory (IRT) based 2-parameter logistic (2-PL) model to develop a population-based t-score (with Gaussian distribution) for symptom presence and intensity (S2 File; S3 File; Table 1). We also performed a differential item functioning (DIF) analysis to identify if people from different subgroups *with the same underlying symptom presence and intensity* have a different probability of providing a certain answer to an item. For example, when an item that only applies to men (e.g., an item addressing erectile dysfunction) is administered to everyone, the score will be consistently different for men than women while the underlying symptom presence and intensity would otherwise be the same. On this basis, we omitted painful menstrual periods (only relevant for women), falls (older patients fell more), loss of control of bladder (more common for women), and seizures (older patients had more seizures). We then repeated confirmatory factor analysis and IRT based 2-PL and found good fit statistics. The included symptoms that measured the same construct were converted into a t-score of PASC symptom presence and intensity (PASC-SPI, S3 File), which was used as our primary outcome. The second factor constituting reduced taste and reduced smell was used as secondary outcome for an unplanned secondary analysis.

## Independent variables

We included sociodemographic, clinical, and mental health variables that may be associated with PASC-SPI based on current thought, evidence, and clinical experience (Table 2).

## Statistical analysis

We used parametric and non-parametric bivariate analyses depending on data distribution to seek factors associated with the long COVID symptom presence and intensity scale, loss of taste, and loss of smell. All variables with a *P*-value of below 0.10 in bivariate analysis were moved to multivariable linear and logistic regression to assess independent associations. We addressed multicollinearity of mental health measures (PHQ-9, GAD-7, and PC-PTSD-5)[32] by running separate models with one mental health measure at the time, selecting the model with the lowest Akaike Information Criterion (AIC) and highest $R^2$ (model with GAD-7 alone) and excluded other mental health measures from the analysis (PHQ-9 and PC-PTSD-5; S2 File). All *P*-values below 0.05 were considered statistically significant.

## Results

The mean age of study subjects was 44 ±16 and 65% were women (Table 3). The majority (84%) reported their COVID-19 diagnosis as confirmed by lab criteria or by a clinician (Table 3).

**Table 1. Item selection for PASC symptom presence and intensity score and results of CFA and IRT based 2-PL analysis.**

| PASC symptom presence and intensity score | Reason for exclusion | Factor loading | R² | Item difficulty (b) | Item discrimination (a) |
|---|---|---|---|---|---|
| **Symptom type** | | | | | |
| Anxiety | Mental health | | | | |
| Unusual behavior or change in personality | Omitted in EFA | | | | |
| **Balance or walking problems** | | 0.75 | 0.56 | -0.12 | 1.9 |
| **Can't move and/or feel one side of body or face** | | 0.70 | 0.49 | 1.8 | 1.8 |
| **Chest pain** | | 0.55 | 0.30 | 0.18 | 0.94 |
| **Constipation** | | 0.57 | 0.32 | 0.92 | 1.1 |
| Depressed mood | Mental health | | | | |
| **Diarrhea** | | 0.58 | 0.34 | 0.83 | 1.2 |
| Painful menstrual periods | DIF (gender) | | | | |
| **Dizziness or lightheadedness** | | 0.77 | 0.59 | -0.60 | 2.0 |
| Erectile dysfunction | Omitted in EFA | | | | |
| **Fainting or blackouts** | | 0.56 | 0.32 | 2.0 | 1.2 |
| Falls | DIF (age) | | | | |
| **Fatigue** | | 0.69 | 0.47 | -1.7 | 1.7 |
| **Fever** | | 0.53 | 0.28 | 2.1 | 1.1 |
| Forgetfulness or "brain fog" | Omitted in CFA | | | | |
| **Hallucinations** | | 0.63 | 0.40 | 2.2 | 1.5 |
| **Headaches** | | 0.71 | 0.50 | -0.53 | 1.7 |
| **Jerking of limbs** | | 0.71 | 0.51 | 0.76 | 1.8 |
| **Joint pain or swelling** | | 0.67 | 0.45 | -0.034 | 1.5 |
| **Loss of appetite** | | 0.68 | 0.47 | 0.58 | 1.6 |
| Loss of control of bladder | DIF (gender) | | | | |
| **Nausea or vomiting** | | 0.71 | 0.50 | 0.60 | 1.7 |
| **Numbness or tingling** | | 0.69 | 0.48 | 0.099 | 1.6 |
| **Pain on breathing** | | 0.67 | 0.44 | 0.91 | 1.4 |
| **Palpitations or heart racing** | | 0.63 | 0.40 | -0.22 | 1.3 |
| **Persistent cough** | | 0.47 | 0.22 | 1.4 | 0.88 |
| **Problems hearing** | | 0.70 | 0.49 | 0.99 | 1.6 |
| **Persistent muscle pain** | | 0.77 | 0.59 | 0.17 | 2.0 |
| **Pain or fatigue after exercise** | | 0.56 | 0.31 | -0.79 | 1.2 |
| **Problems passing urine** | | 0.76 | 0.58 | 1.5 | 2.2 |
| **Problems seeing** | | 0.70 | 0.48 | 0.51 | 1.7 |
| **Problems swallowing** | | 0.68 | 0.46 | 1.3 | 1.7 |
| Reduced smell | Omitted in EFA | | | | |
| Reduced taste | Omitted in EFA | | | | |
| **Ringing in ears** | | 0.65 | 0.43 | 0.33 | 1.4 |
| Seizures | DIF (age) | | | | |
| **Shortness of breath** | | 0.53 | 0.28 | -0.69 | 0.94 |
| **Skin rash or changes in color** | | 0.62 | 0.39 | 1.0 | 1.4 |
| **Slowness of movement** | | 0.72 | 0.52 | 0.15 | 1.7 |
| Sleeping more | Omitted in CFA | | | | |
| **Sleeping less, or difficulty falling asleep** | | 0.67 | 0.45 | -0.30 | 1.4 |
| **Stiffness of muscles** | | 0.77 | 0.60 | 0.17 | 2.0 |
| **Tremors** | | 0.73 | 0.53 | 0.71 | 1.8 |
| **Trouble in concentrating** | | 0.74 | 0.54 | -1.0 | 1.9 |
| **Weakness in arms or legs** | | 0.76 | 0.58 | -0.13 | 2.0 |

*(Continued)*

**Table 1.**  (Continued)

| PASC symptom presence and intensity score | Reason for exclusion | Factor loading | $R^2$ | Item difficulty (b) | Item discrimination (a) |
|---|---|---|---|---|---|
| RMSEA (CFA) = 0.035, Cronbach alpha = 0.92 | | | | | |

$R^2$ = the percentage of variance of each item that is explained by the latent factor (symptom intensity). The higher $R^2$, the better the item is at measuring symptom intensity.

Item discrimination (a) = the ability of an item to identify different levels of symptom intensity among patients

Item difficulty (b) = the level of symptom intensity required to achieve a 0.5 probability of having a certain symptom

Accounting for potential confounding variables, we found that greater combined PASC-SPI was associated with greater symptoms of anxiety, a longer duration of symptoms during the initial COVID-19 infection, and hypercholesterolemia. Lower combined PASC-SPI was associated with older age, self-reported 1–3 units of alcohol per week compared to none, and a self-reported clinician confirmation COVID-19 diagnosis (Table 4). Symptoms of anxiety accounted for a considerably higher proportion of variation in PASC-SPI (semi-partial $R^2$) than other variables.

Accounting for potential confounding between variables, we found that there were no factors associated with reduced taste or reduced smell.

## Discussion

Treatment of PASC is a burgeoning field as researchers strive to define the condition, identify underlying mechanisms, and establish evidence-based interventions. Despite fatigue, neuro-cognitive dysfunction, and dizziness being the most commonly reported symptoms among our cohort, it was the presence of anxiety that was most closely associated with a higher proportion of variation in PASC-SPI. This underscores the need for a multifaceted treatment approach that addresses not only the biological impact of the virus but also the psychological and social factors contributing to the patient experience of PASC. Routine administration of a comprehensive and standardized survey based on the WHO's Global COVID-19 Clinical Platform Case Report Form for Post COVID Condition in a PASC unit in an urban US city, combined with standardized mental health screening of symptoms of anxiety and depression, provided an opportunity to analyze variables associated with PASC-SPI.

## Limitations

This study has several limitations. First, this is a retrospective study using an established database of symptoms. PASC is loosely defined (over 200 symptoms reported) and others might include or exclude symptoms we collected. However, we used sophisticated statistical techniques in an attempt to combine all the potentially relevant PASC symptoms. Similarly, we used the same statistical techniques to convert the binary measure of symptoms (present or absent) into a single intensity score. In that same light, we assessed the presence of symptoms rather than an ordinal scale that measures the intensity of each symptom. Although an ordinal scale for each symptom might provide a more accurate symptom intensity scale, the factor structure is likely to be similar. We considered this analysis a useful step towards determining which symptoms contribute most to the core group of symptoms considered PASC, and identifying modifiable factors associated with the presence and intensity of symptoms that can be targeted in treatment strategies and future research. Third, this study included only persons referred to a dedicated PASC clinic who were screened by a triage nurse. Our results may not apply to PASC in different populations. However, studies performed in different settings and among different populations confirm a similar correlation between

**Table 2. Summary of the measured independent sociodemographic, clinical, and mental health variables.**

| Sociodemo-graphic variables | Description | Clinical variables | Description | Mental health variables | Description |
|---|---|---|---|---|---|
| Age | Years | Body mass index (BMI) | Kg/M² | 7-item version of the Generalized Anxiety Disorder (GAD-7) | The GAD-7 scale ranges from 0 to 21, with a higher score indicating more severe symptoms of anxiety. Patients are asked how often they have been bothered by certain problems over the last 2 weeks (not at all, several days, more than half the days, or nearly every day), and examples of problems include: "Feeling nervous, anxious or on edge" or "Not being able to stop or control worrying". |
| Gender | Male or female | Smoking status | Current, former, and never | | |
| | | Marijuana use | Yes/ No | | |
| Race | Caucasian/white, African-American/ black, Asian, and other | Self-reported alcohol use per week | None, 1–3 units, 3–6 units, and >6 units | | |
| Level of education | Less than high school, high school or GED, and college or professional | Pre-existing conditions | Asthma, auto-immune, chronic lung disease, diabetes, fibromyalgia, high cholesterol, high blood pressure, and migraines | | |
| | | COVID-19 vaccination | Yes/ No | 9-item version of the Patient Health Questionnaire (PHQ-9) | The PHQ-9 scale ranges from 0 to 27, with a higher score indicating more severe symptoms of depression and has the same set-up as GAD-7. Examples of problems include: "Little interest or pleasure in doing things" or "Feeling down, depressed or hopeless". |
| | | Confirmed COVID-19 diagnosis | Yes/ No | | |
| | | Symptom duration | 1-5 days, 5–10 days, 10–20 days, and 20–30 days | | |
| | | Care type received during COVID-19 infection | Admitted at hospital, emergency department only, outpatient UC, and home | | |
| | | Self-reported COVID-19 severity | No symptoms, mild: I had symptoms (e.g., cough, fever, diarrhea) but did not get short of breath or require oxygen, moderate: I felt short of breath or had chest imaging showing pneumonia, but did not require oxygen or hospitalization, severe: I had COVID-19 pneumonia requiring oxygen and/or admission to a hospital, and critical: I required mechanical ventilation or maximum respiratory support in an intensive care unit, or had multi-organ damage. | 5-item Primary Care PTSD Screen for DSM-5 (PC-PTSD-5) | The PC-PTSD-5 scale ranges from 0 to 5, with a higher score indicating a higher likelihood of post-traumatic stress disorder. Examples of questions include "In the past month, have you been constantly on guard, watchful, or easily startled?" or "In the past month, have you felt numb or detached from people, activities, or your surroundings?" and patients answer with yes or no. |

mental health and symptom intensity[8,17,33]. Fourth, participation bias was likely among this study population since symptom severity and levels of distress may both influence care seeking behavior. The findings of our study apply best to people with sufficient concerns or socioeconomic resources to seek care and may not apply more broadly to the population of patients with PASC, many of whom may be accommodating symptoms or not be aware of the availability of PASC specific care. Lastly, symptoms of falls (older patients fell more), and loss of control of bladder (more common for women) were omitted due to the risk of spurious associations in statistical models due to anticipated variations in answers among certain subgroups. However, it is possible that these aspects of PASC, which may be manifestations of pathophysiologies such as dysautonomia and small fiber neuropathy, are relevant for certain patient subgroups.

## Variables associated with long COVID symptom presence and intensity scale

The finding that greater PASC-SPI was strongly associated with greater symptoms of anxiety and longer duration of acute COVID-19 infection points to the importance of anticipating and addressing the psychosocial aspects of PASC. The correlation with symptoms of anxiety is

**Table 3. Demographics of cohort.**

| Variables | Value* |
|---|---|
| N | 249 |
| Age | 44 ± 16 |
| Female | 161 (65%) |
| Body Mass Index | 30 ± 8.5 |
| Smoking status | |
| Current | 7 (2.8%) |
| Former | 75 (30%) |
| Never | 167 (67%) |
| Marijuana use | 21 (8.4%) |
| Alcohol use (per week) | |
| None | 142 (57%) |
| 1-3 | 66 (27%) |
| 3-6 | 27 (11%) |
| >6 | 14 (5.6%) |
| Race | |
| White | 196 (79%) |
| Black/African-American | 9 (3.6%) |
| Asian | 9 (3.6%) |
| Other | 35 (14%) |
| Level of education | |
| Less than high school | 5 (2.0%) |
| High school or GED | 50 (20%) |
| College or professional | 194 (78%) |
| Pre-existing problems | |
| Asthma | 51 (20%) |
| Auto-immune | 33 (13%) |
| Chronic lung disease | 4 (1.6%) |
| Diabetes | 23 (9.2%) |
| Fibromyalgia | 20 (8.0%) |
| High cholestrol | 42 (17%) |
| High blood pressure | 53 (21%) |
| Migraines | 42 (17%) |
| COVID vaccine | 205 (82%) |
| COVID diagnosis | 209 (84%) |
| Symptom duration (days) | |
| 1-5 | 27 (11%) |
| 5-10 | 49 (20%) |
| 10-20 | 81 (33%) |
| 20-30 | 92 (37%) |
| Care type | |
| Admitted at hospital | 23 (9.2%) |
| Emergency department only | 39 (16%) |
| Outpatient UC | 49 (20%) |
| Home | 138 (55%) |
| COVID severity | |
| No symptoms | 7 (2.8%) |
| Mild | 89 (36%) |

*(Continued)*

**Table 3.** (Continued)

| Variables | Value* |
|---|---|
| Moderate | 123 (49%) |
| Severe | 21 (8.4%) |
| Critical | 9 (3.6%) |
| Loss of taste | 81 (33%) |
| Loss of smell | 77 (31%) |
| Employment during COVID | |
| No change | 65 (26%) |
| Reduced hours | 137 (55%) |
| Lost job | 47 (19%) |
| MoCA | 25 ± 3.2 |
| GAD | 6 (2–11) |
| PHQ | 3 (1–14) |
| PTSD | 1.5 (0–3) |
| Long COVID symptom presence and intensity scale (t-score) | 50 ± 9.6 |

*Value is displayed as median with interquartile range for continuous non-parametric variables, as mean with standard deviation for continuous variables with normal distribution, and as number with percentage for categorial variables.

likely bidirectional and could be related to manifestations of COVID-related central nervous system pathophysiology, a product of the stress associated with infection and the pandemic, or pre-morbid traits or tendencies. The social health aspects may include financial stress related to costs of hospitalization[34,35], the need to ration food, heat, housing, and medication[35], reduced income related to reduced hours at work or inability to work[36], and prolonged isolation[37]. It is important to recognize that symptoms of anxiety might stem directly from neuroinflammatory disruptions. Multiple studies have demonstrated decreased serum serotonin, catecholamines and their metabolites in PASC, and autonomic testing in patients with post-infectious ME/CFS indicated heightened sympathetic and reduced parasympathetic activity[9,38–40]. In the absence of a biomarker to assess the severity or impact of these physiological alterations, clinicians treating PASC should proactively anticipate and address feelings of distress as a strategy to alleviate physical symptoms.

Our findings are in line with a prospective cohort study that studied the frequency of 37 symptoms among patients diagnosed with PASC[8] and found that greater symptoms of anxiety and depression were associated with almost all other symptoms, especially non-specific symptoms such as brain fog, fatigue, and dizziness. In contrast, loss of taste or smell was primarily present in the subgroup with the lowest frequency of most other symptoms. Similarly, in a cohort of nearly 55,000 patients, threshold levels of distress and loneliness prior to COVID-19 infection were associated with diagnosis of PASC, and patients with a threshold level of distress experienced more PASC symptoms and more impairment of daily living[17]. A study in the UK compared adolescents (11–17 years old) with PCR confirmed infection and normal PCR test and found that both groups experienced persistent symptoms 3 months after testing (30% and 19%, respectively)[41]. Using a latent class analysis, they found that worse physical and mental health assessed at the time of testing was associated with more PASC symptoms and worse mental health 3 months after testing, independent of evidence of actual infection. Additionally, a systematic review found relatively high pooled prevalence of symptoms of depression and anxiety

**Table 4. Multivariable linear regression analysis of factors associated with the PASC symptom presence and intensity scale (t-score).**

| Variables | Regression Coefficient (95% Confidence Interval) | P-value | VIF | Semi-partial R² |
|---|---|---|---|---|
| Age | -0.76 (-0.15 to -0.0053) | **0.035** | 1.2 | 0.018 |
| BMI | 0.036 (-0.14 to 0.13) | 0.96 | 1.3 | <0.0001 |
| GAD | 0.48 (0.31 to 0.64) | **<0.001** | 1.1 | 0.087 |
| Alcohol (per week) | | | | 0.010 |
| None | *Reference value* | | | |
| 1-3 | -2.9 (-5.4 to -0.35) | **0.026** | 1.2 | |
| 3-6 | -2.3 (-5.8 to 1.2) | 0.19 | 1.1 | |
| >6 | -2.3 (-7.0 to 2.3) | 0.33 | 1.1 | |
| Symptom duration (days) | | | | 0.033 |
| 1-5 | *Reference value* | | | |
| 5-10 | 6.0 (1.9 to 10) | **0.004** | 2.5 | |
| 10-20 | 4.9 (0.91 to 8.8) | **0.016** | 3.3 | |
| 20-30 | 7.6 (3.5 to 12) | **<0.001** | 3.8 | |
| COVID severity | | | | <0.0001 |
| No symptoms | *Reference value* | | | |
| Mild | 0.93 (-5.9 to 7.8) | 0.79 | 10 | |
| Moderate | 3.3 (-3.6 to 10) | 0.35 | 11 | |
| Severe | 0.22 (-7.6 to 8.1) | 0.96 | 4.5 | |
| Critical | -2.0 (-11 to 7.0) | 0.66 | 2.7 | |
| Gender | | | | 0.011 |
| Male | *Reference value* | | | |
| Female | 2.1 (-0.29 to 4.5) | 0.086 | 1.2 | |
| COVID diagnosis | | | | 0.015 |
| No | *Reference value* | | | |
| Yes | -3.1 (-6.1 to -0.19) | **0.037** | 1.1 | |
| Pre-existing problems | | | | |
| Asthma | 1.6 (-1.2 to 4.4) | 0.26 | 1.2 | 0.0034 |
| Auto-immune | -0.54 (-3.8 to 2.7) | 0.75 | 1.2 | <0.0001 |
| Chronic lung disease | 5.3 (-3.4 to 14) | 0.23 | 1.1 | 0.0031 |
| Diabetes | 3.7 (-0.41 to 7.8) | 0.078 | 1.4 | 0.0063 |
| High cholesterol | 3.4 (0.43 to 6.4) | **0.025** | 1.2 | 0.018 |
| Migraine | -0.31 (-3.3 to 2.7) | 0.84 | 1.2 | 0.0001 |

Model fit statistics: $R^2$=0.32, adjusted $R^2$=0.26, Akaike Information Criterion=1779

(EQ-5D; 38% [19% to 58%]), fatigue (64% [54% to 73%]), low quality of life (EQ-VAS; 59% [42% to 74%]), and sleep disturbance (47% [7% to 89%]) [42] among people diagnosed with PASC. There is also evidence that distress correlates with both the diagnosis and severity of (respiratory tract) infections[43–46] as well as postinfectious fatigue syndromes[47].

The consistent, notable associations of mental health and PASC symptoms underline the importance of incorporating mental health into care pathways. Interventions that address mental health can alleviate physical symptoms among people with a variety of conditions[48–51]. A recent multicenter randomized controlled trial found that cognitive behavioral therapy resulted in significant improvement of fatigue (Checklist Individual Strength fatigue severity subscale) up to 6 months after therapy among patients with severe fatigue 3–12 months after COVID-19 infection[52]. The REGAIN study, a pragmatic multicenter

RCT offering an online supervised group rehabilitative program with a focus on exercise and psychological support, demonstrated efficacy in improving health-related quality of life[53]. This indicates that while integrated rehabilitative programs for PASC may not cure the illness, they can assist patients in coping with their symptoms and enhancing their functional capacity. Evaluating mental health symptoms beyond the simplistic binary of presence or absence, especially in the context of complex conditions like depression and anxiety, may offer a more nuanced understanding. This approach acknowledges the complex nature of these diagnoses, where distinguishing between cognitive-emotional aspects, such as feelings of worthlessness, rumination or worry, and physiological manifestations like fatigue and limited concentration, is crucial. Such a nuanced assessment recognizes the continuum of mental health experiences, potentially offering more targeted insights into how alleviating specific symptoms of depression or anxiety could also mitigate PASC symptoms. Categorization of mental health can also reinforce mental health stigma. Regrettably, many patients experience stigmatization or trivialization of their symptoms by clinicians, friends, and family[54,55] and it is our hope that our findings will not contribute to this.

## Variables associated with loss of taste and loss of smell

While many of the diffuse symptoms of PASC are constitutional or non-specific, anosmia and aguesia are features that can be directly attributed to COVID-19 infection. The observation that there were no factors associated with either reduced taste or reduced smell is consistent with the evidence that anosmia and ageusia are neither pathognomonic for PASC nor clearly associated with other symptom clusters[56–59]. Utilizing anosmia and ageusia as more as an "objective" measure of PASC physiology is not helpful, as they are independent of the other symptoms and confounders. One study found that loss of taste or smell was primarily present in the subgroup with the lowest frequency of most other symptoms[8]. The other found that loss of smell was largely independent from other PASC symptoms such as fatigue or cognitive deficits, and that the subgroup characterized by olfactory dysfunction was associated with lower levels of anxiety, depression, and stress (Patient Health Questionnaire)[33]. On the other hand, there are some studies that have correlated loss of taste or smell with disruption of daily living and feelings of distress[60,61]. One of these studies was a qualitative study among users of a Facebook group for peers with loss of taste and smell, potentially representing a population with higher levels of distress regarding their symptoms. A cross-sectional study among 104 healthcare workers found that loss of taste was associated with higher levels of distress (0–10 Likert scale), but that neither loss of taste or smell were associated with symptoms of depression and anxiety (Hospital Anxiety and Depression Scale)[62].

## Conclusions

We analyzed many factors and found that age, hypercholesterolemia, duration of symptoms during the initial COVID-19 infection, self-reported clinician confirmation COVID-19 diagnosis, self-reported alcohol use, and symptoms of anxiety were associated with the presence and intensity of symptoms ascribed to PASC. Symptoms of anxiety accounted for the largest proportion of the variation in PASC symptom presence and intensity levels. Future studies might elaborate on the relationship of psychosocial factors with PASC symptom intensity as well as the best ways to incorporate these aspects into comprehensive, whole person care for patients diagnosed with PASC regardless of the underlying pathophysiology.

## Supporting information

**S1 File. The 46 symptoms ascribed to PASC included in the factor analysis.** The 46 included symptoms that were derived from a comprehensive review of systems modeled on the WHO's Global COVID-19 Clinical Platform Case Report Form for Post COVID Condition.
(DOCX)

**S2 File. Details of the statistical analysis.** A more elaborate explanation of the statistical analyses used.
(DOCX)

**S3 File. Conversion table for the PASC symptom presence and intensity score.** The formula that can be used to generate the PASC symptom presence and intensity scores.
(DOCX)

**S4 File. Deidentified data.** All data used to generate the findings of this study.
(CSV)

## Acknowledgement

We would like to thank Peyman Jafari for his help with performing the analyses.

This study was conducted at the Department of Internal Medicine & Department of Population Health, Dell Medical School, The University of Texas at Austin.

W. Michael Brode receives consulting fees from Roon Healthcare as a paid consultant and content expert and has stock or stock options at Roon Healthcare, payment or honoraria for lectures, presentations, speakers' bureaus, manuscript writing or educational events at Texas Health and Human Services (honorarium paid to University of Texas at Austin), HNI Health Value Institute (honorarium paid directly as individual), and Society of Hospital Medicine (honorarium paid directly as individual), and support for attending meetings and/or travel from Society of Hospital Medicine for travel and lodging for the 2022 Meeting.

## Author contributions

**Conceptualization:** Teun Teunis, David Ring, W. Michael Brode.

**Data curation:** Niels Brinkman.

**Formal analysis:** Niels Brinkman, Seung Choi.

**Funding acquisition:** W. Michael Brode.

**Investigation:** Niels Brinkman, Teun Teunis, W. Michael Brode.

**Methodology:** Niels Brinkman, Seung Choi.

**Project administration:** W. Michael Brode.

**Resources:** Teun Teunis, Seung Choi, W. Michael Brode.

**Software:** Seung Choi.

**Supervision:** Teun Teunis, Seung Choi, David Ring, W. Michael Brode.

**Validation:** Seung Choi.

**Visualization:** Niels Brinkman, Teun Teunis, Seung Choi, David Ring, W. Michael Brode.

**Writing – original draft:** Niels Brinkman.

**Writing – review & editing:** Teun Teunis, David Ring, W. Michael Brode.

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
