## [Decision Letter · Decision Letter 0]

22 Jan 2025

PONE-D-24-51383Factors associated with the presence and intensity of ongoing symptoms in Long COVIDPLOS ONE

Dear Dr. Ring,

Thank you for submitting your manuscript to PLOS ONE. After careful consideration, we feel that it has merit but does not fully meet PLOS ONE’s publication criteria as it currently stands. Therefore, we invite you to submit a revised version of the manuscript that addresses the points raised during the review process. Unfortunatekly, I could find only one reviewer for your manuscript.As suggested by this reviewer, please describe the purpose of your study. Please submit your revised manuscript by Mar 08 2025 11:59PM. If you will need more time than this to complete your revisions, please reply to this message or contact the journal office at plosone@plos.org. Please include the following items when submitting your revised manuscript:

We look forward to receiving your revised manuscript.

Kind regards,

Etsuro Ito, Ph.D.

Academic Editor

PLOS ONE

2. We note that your Data Availability Statement is currently as follows: [All relevant data are within the manuscript and its Supporting Information files.] Please confirm at this time whether or not your submission contains all raw data required to replicate the results of your study. Authors must share the “minimal data set” for their submission. PLOS defines the minimal data set to consist of the data required to replicate all study findings reported in the article, as well as related metadata and methods (https://journals.plos.org/plosone/s/data-availability#loc-minimal-data-set-definition).

4. Please include a copy of Table 1,2,3,4 which you refer to in your text on page 7 and 8.

Reviewers' comments:

Reviewer's Responses to Questions

**Comments to the Author**

1. Is the manuscript technically sound, and do the data support the conclusions?

Reviewer #1: Partly

2. Has the statistical analysis been performed appropriately and rigorously? 

Reviewer #1: Yes

3. Have the authors made all data underlying the findings in their manuscript fully available?

Reviewer #1: Yes

4. Is the manuscript presented in an intelligible fashion and written in standard English?

Reviewer #1: Yes

5. Review Comments to the Author

Reviewer #1: The article addresses a relevant and globally significant public health issue. However, I believe some parts could be adjusted.

The title and the short title seem to belong to different articles, as they are not consistent with each other. For this reason, it is unclear what the focus of the article will be.

Regarding the objective, it could be written in a more specific way that clearly reflects the purpose of the study, as it is currently confusing what the study aims to achieve. This is compounded by the fact that the titles differ.

I think it is important that the introduction clearly states the purpose of the study, as it mentions the identification of factors associated with the presence and intensity of ongoing symptoms in Long COVID, but then it discusses anxiety and depression (are these the identified factors?). This is somewhat confusing. Similarly, in the results section, anxiety factors and symptoms are mentioned first.

Although the discussion mentions and clarifies associated factors, the relationship with anxiety remains unclear.

Finally, in the conclusion, the study does not seem to answer the stated objective (to identify associated factors), as it only discusses anxiety factors and symptoms as the sole associated factors.

6. PLOS authors have the option to publish the peer review history of their article (what does this mean?). If published, this will include your full peer review and any attached files.

Reviewer #1: **Yes: **Diana Carolina Zona Rubio

---

## [Author Response · Author response to Decision Letter 1]

8 Feb 2025

Please see file attached (response to reviewer comments) for more the more comprehensive format of our responses. A quick summary of our responses below:

Answer first comment: Please see edits throughout.

Answer second comment: Curious that you find them unrelated? We would appreciate some elaboration if possible. The title and subtitle should reflect the applicable knowledge from the experiment, and—in our view—they do. The subtitle refers to the most important findings related to the factors associated with symptom presence and intensity ascribed to PASC (symptoms of anxiety was the strongest correlate).

Answer third comment: We’ve added a sentence elaborating on the study’s purpose (line 129-131):

“The purpose of this study was to identify which factors, including psychosocial aspects of health, are associated with symptom presence and intensity ascribed to PASC.”

Answer fourth comment: Please see edits in the manuscript to further elaborate on the study purpose and the emphasis on mental health.

In the results, symptoms of anxiety are mentioned first as they turned out to be the strongest correlate of symptom presence and intensity ascribed to PASC. We chose to describe the most important / strongest correlates first as the order would otherwise be random. The order of mentioning the correlates does otherwise not matter. We’ve also added the following references in the edits:

Castanares-Zapatero D, Chalon P, Kohn L, Dauvrin M, Detollenaere J, Maertens de Noordhout C, et al. Pathophysiology and mechanism of long COVID: a comprehensive review. Ann Med 2022;54:1473. https://doi.org/10.1080/07853890.2022.2076901.

Answer fifth comment: Acknowledged. Please see edits in the discussion.

Answer sixth comment: Acknowledged. Please see edits in the conclusion.

---

## [Editor Report · Decision Letter 1]

11 Feb 2025

Factors associated with the presence and intensity of ongoing symptoms in Long COVID

PONE-D-24-51383R1

Dear Dr. Ring,

We’re pleased to inform you that your manuscript has been judged scientifically suitable for publication and will be formally accepted for publication once it meets all outstanding technical requirements.

Kind regards,

Etsuro Ito, Ph.D.

Academic Editor

PLOS ONE

---

## [Editor Report · Acceptance letter]

PONE-D-24-51383R1

PLOS ONE

Dear Dr. Ring,

I'm pleased to inform you that your manuscript has been deemed suitable for publication in PLOS ONE. Congratulations! Your manuscript is now being handed over to our production team.

Kind regards,

on behalf of

Prof. Etsuro Ito

Academic Editor

PLOS ONE